# When Agents Talk Back: Rebellious Explanations

**Ben Wright**[1]  and  **Mark Roberts**[2]  and  **David W. Aha**[2]  and  **Ben Brumback**[2]

[1]NRC Postdoctoral Fellow, Navy Center for Applied Research and Artificial Intelligence
[2]Navy Center for Applied Research in Artificial Intelligence
Naval Research Laboratory
{benjamin.wright.ctr, mark.roberts, david.aha, benjamin.brumback}@nrl.navy.mil

## Abstract

As the area of Explainable AI (XAI), and Explainable AI Planning (XAIP), matures, the ability for agents to generate and curate explanations will likewise grow. We propose a new challenge area in the form of rebellious and deceptive explanations. We discuss how these explanations might be generated and then briefly discuss evaluation criteria.

## 1 Introduction

Explanations as a research area in AI (XAI) has been around for several decades (Clancey 1986; Buchanan and Shortliffe 1984; Craven 1996; Wick and Thompson 1992; Doyle et al. 2003; Sørmo et al. 2005; Weston et al. 2015; Miller 2018). It has additionally gained momentum recently as evidenced by the increasing number of workshops and special tracks covering it in various conferences (e.g., VIS-xAI, FEAP-AI4Fin, XAIP, XAI, OXAI, MAKE-eXAI, ICCBR-19 Focus area).

While still growing in use, there have been some approaches to formalizing XAI. DARPA (2016) stated that anything calling itself XAI should address the following questions:

- Why did the agent do that and not something else?

- When does the agent succeed and when does it fail?

- When can I trust the agent?

However, less thought out is the idea of explanations that are *deceptive* or *rebellious* in nature. These forms of explanation can be an entirely new area of discussion and use for certain autonomous agents.

The study of deception and rebellion are both rich fields, and many aspects of both that are studied in civilian and military capacities. For example, the area of deception detection works on finding ways to detect inconsistencies (Thomas and Biros 2011; Kott et al. 2011; Biros et al. 2005). Isaac and Bridewell (2017) discuss a number of ways why deception is an important topic for autonomous agents.

Studies of rebellion and resistance have investigated how, why, when it does, and doesn't, happen (Martí and Fernández 2013; Pershing 2003b). The use of both has also been studied (Reed 2016; Anderson et al. 2004; Kott and Ownby 2005; Keller et al. 2015; Mourougayane and Srikanth 2015).

The idea of pairing deception and rebellion with explanations may not be intuitive initially. However, in addition to being areas of rich study, deception and rebellion offer key conditions that are of interest to agent reasoning. Predominately, it requires multiple actors (i.e., An actor *deceives* another actor, or an actor *rebels* against a coordinator). Additionally, there needs to be some sort of conflict or misalignment between the actors. Either something needs to be in contention for an actor to rebel, or something needs to be in conflict for the actor to use deception. b Rebellion in agents has been a growing area of interest (Aha and Coman 2017; Coman and Aha 2018; Dannenhauer et al. 2018; Boggs et al. 2018; Coman and Aha 2017). This area is focused on finding models in which agents can *rebel* from directives given in certain circumstances. This can include having more up-to-date knowledge that would affect the plan, finding opportunities to exploit but may be off-mission, or solving problems or roadblocks before they become an issue even if it is off-mission. (Aha and Coman 2017) discuss three ways in which rebellion could manifest in agents. The *expression* of a rebellion can consist of either an explicit or implicit act. The *focus* is either inward or outward facing. Lastly, the *interaction initiation* can either be reactive or proactive.

Deception in agents has been progressing over the last decade, with many discussions on formalizing deception. The majority of this formalism is on the topic of lying (Van Ditmarsch et al. 2012; Sakama et al. 2011; Van Ditmarsch 2014). There has also been inroads for more encompassing deception as described by (Sakama 2015) and (Sakama and Caminada 2010). Of interest here, (Sakama, Caminada, and Herzig 2014) defined *Quantitative & Qualitative Maxims for Dishonesty* as the following maxims:

1. Lie, Bullshit (BS), or withhold information as little as possible to achieve your objective.

2. Never lie if you can achieve your objective by BS.

3. Never lie nor BS if you can achieve your objective by withholding Information.

4. Never lie, BS, nor withhold information if you can achieve your objective with a half-truth.

A particular topic that has received attention is deceptive, or dishonest, agents in negotiations (Sakama et al. 2011; Nguyen et al. 2011; Zlotkin and Rosenschein 1991).

With these concepts in mind, we will pursue research to answer the following:

*What kind of reasoning models are required to generate explanations of a deceptive or rebellious nature?*

## 2 Related Work

Research into rebellion and deception have been studied for decades in other fields. We summarize some of these studies in this section, as they will form the foundation and context of our approach.

### 2.1 Dissent

Dissent assumes a challenge to some system of power or belief (Martin 2008). The author goes further to differentiate between dissent and rebellion. Martin claims that a basic distinction between dissent and rebellion is that dissenters still believe in the process that has been established whereas rebels do not.

### 2.2 Whistle-blowing

A number of real-world studies on dissent and whistle-blowing in organizations showcase the importance of this topic and area (Hamid et al. 2015; Kenny et al. 2018; Martin 2008; Pershing 2003a; Martin and Rifkin 2004). Many of these discuss the outcomes and results towards both the actor and the agent in these situations.

Martin and Rifkin (2004) discuss organizational responses to whistle-blowing and dissent. Manager reprisals may include building a damaging record towards them, threats against them, isolation, prosecution, or setting them up for failure at their job. In a number of instances, it seems that giving dissent, or becoming a whistle-blower, can quickly turn into an adversarial or antagonistic situation.

### 2.3 Cynicism

Closely tied to dissent is the idea of cynicism. Mantere and Martinsuo (2001) defines it as:

(1) a belief that there is a gap between desired and observed organizational identity; (2) a negative affect toward the organization or organizational change (strategy); and (3) tendencies to disparaging and/or critical behaviors toward the organization that are consistent with those beliefs and affect.

They additionally give examples of cynicism as: pessimism, emotional/narrative expressions or outbursts, frustration, irony, accusations, neglect, negative coping behaviors and expressions, and aggression.

### 2.4 Subversion

Observed actions of resistance to change have been studied in a number of ways. Once such study (Ybema and Horvers 2017) noted that there was a tendency towards "informal or mundane" types of resistance compared to an open, direct, and explicit form of objection. When faced with managerial initiatives, the following expressions of resistance were noted: "careful carelessness", humor, cynicism and skepticism, nostalgic talk (i.e., the "Good ol' Days"), alternative articulations of self-hood, and simulation of productivity.

While these expressions of resistance were present, workers were also camouflaging this dissent with a good-humored appearance. It was noted that hiding dissent allowed for behind-the-scenes inaction to stifle any new directives and also avoided many conversations that workers deemed futile. Similar studies include (Ewick and Silbey 2003; McKay et al. 2013; Reissner 2011).

Along with giving a number of examples of resistance, Martí and Fernández (2013) defined a few different kinds of resistance. Is the resistance individual or collective in nature? Covert or overt? Mundane or heroic?

While overt forms of resistance usually did not work, the *stories* of that resistance remained and were used later to continue forms of resistance.

### 2.5 Observational Deception Studies

There have been several studies on how deception in society works. A very good resource is (Whiten and Byrne 1988) which defines a number of deceptive actions observed among primates in the wild. They are categorized as either *concealment, distraction, use of a tool,* or *use of another agent.* Examples include hiding things from sight, looking at things to have others avoid looking at something else, and getting other agents to take the blame for something. In addition to primates, there have been deceptive studies for cephalopods (Brown et al. 2012) and dolphins (Hill et al. 2018).

In studies of deception in warfare (Reed 2016), there have been noticeable benefits to using deception. This includes minimizing the amount of resources for a task or ensuring the opponents mis-allocate their resources. Also, most of the example deceptive acts required an extended duration of time to be executed and prove successful. This included preparations for and performing the deceptive actions.

## 3 Example (Ecological Disaster)

A good case study from (Dunin-Keplicz and Verbrugge 2011) describes an ecological disaster scenario. We present it now to provide context for explanations discussed in Section 4. In the ecological disaster scenario, two poisons have breached containment in a large area. Each poison is highly dangerous on its own. However, when combined they could be explosive. Robots on the ground have the ability to neutralize specific poisons in small areas. An unmanned air vehicle (UAV) can survey large areas to identify high concentrations of poison. A coordinator can issue commands to either the UAV or the ground robots.

A robot can gauge the type and quantity of poisons in the small area around it, and can move across the ground. The UAV can scan larger areas to identify large concentrations of individual poisons, though it reports only the highest quantity. Additionally, the UAV cannot scan areas that are obscured by ceilings or coverings. The coordinator receives the

information from the UAV and the Robots, but does not have a view of their own.

Examples of rebellion here could be robots not following commands to enter areas that would explode or the UAV deciding to survey a different area to help robots on the ground compared to an area that the coordinator instructed the UAV to survey.

# 4 Example Deceptive & Rebellious Explanations

Let us now consider explanations that are rebellious or deceptive within the context given in Section 3. Some of the recurring reasons for generating these explanations include "makes it simpler" or "avoids a larger conversation". These can limit conversations but can also cause misinterpretations. It also has the ability to avoid conversation bottlenecks, letting agents continue to perform other actions.

## 4.1 An Explanation with Lying

During a disaster, it may come to pass that an agent with the ability to administer first-aid or medical attention, perhaps they are equipped with air masks or oxygen, encounters a victim who will only let them near if they do not inform any law enforcement of their position. In this instance, the agent would administer the first aid or medical treatment and, when asked by the Coordinator to explain their activities, would say they are performing a different activity or in a different location.

## 4.2 An Explanation that Withholds Information

A robot could be traversing an area trying to help people evacuate or administer aid and they have informed the Coordinator they are performing this task. However, suppose there is a person who wishes to remain anonymous or would refuse help. In this case, the robot could explain its actions but leave out details that would identify the person in later reports or debriefs. This would help save the person and increase that victim's trust in the robot to help them out of the area.

## 4.3 An Explanation that is only a Half-Truth.

A version of an explanation with a half-truth could be as follows: a medical agent has found and is administering aid to a victim, however the victim is too far gone. Keeping the victim calm with a half-truth or "white lie" explanation would be beneficial to the victim.

## 4.4 An Explanation that is a Protest

An example of a protest-based explanation could come from the following contingency. An exploratory or search agent has encountered an area of the environment that it deems too hazardous to continue in. The Coordinator asks why it isn't moving forward anymore. The agent responds with,"I will not move forward until the hazardous area has been secured and is safe to pass through."

## 4.5 An Explanation that is Cynical

As discussed in Section 2.3, cynicism is a bit odd. It is usually used as a form of soft-resistance. The agent still performs the action or command, but may not do it optimally. An example could be that the Coordinator assigns a robot that has a full amount of neutralizing and medical equipment to survey an area. This might take a while for the robot to execute, so the Coordinator might ask why progress is slow, and the explanation could be "If I could fly this would go much quicker." Alternatively, asking to release all of its equipment so that it can be lighter to perform the survey is another example.

## 4.6 An Explanation with Disobedience

For this instance, the agent is in some sort of situation in which it will not continue an objective given by the Coordinator. Perhaps the Coordinator has tasked an agent with neutralizing an area under a fallen concrete roof. However the agent has noticed a victim in the area being treated by another agent. In that instance the agent could respond to the Coordinator, "I will not neutralize that area, there is a victim in the vicinity. Please assign me a different objective."

# 5 Enablers of Rebellious Explanations

In order to generate possible deceptive explanations as suggested above an agent would require a few things in its models to properly generate a model. An agent would require an internal model of the domain so that it can reason about possible actions, tasks, and goals. It would also require an internal model of the *external agent's* domain model. This is required so that when generating the "deceptive" aspect, it can be validated against the presumed model of that agent. In addition to these models, a few conditions should be met as well. Notably, a discrepancy must be noticed between the external agent's model and the internal model in relation to the query asked. There needs to be a specific condition or contingency in which the truth would not maximize an overall objective benefit for the agent in question. Likewise, rebellious explanations require similar things such as internal models for both the domain and objectives along with noticing a discrepancy between the objectives, the domain, and the agent's internal model.

Of great interest is the work in model reconciliation (Chakraborti et al. 2017). This is focused on maintaining (at least) two separate models, an internal one to the agent, and an external one for someone interacting with the agent. The idea is for the agent to *reconcile* any differences between these two versions to formulate an explanation. This approach is promising in regards to expanding it towards rebel or deceptive tasks in explanation.

In the case of either deceptive or rebellious explanations, a discrepancy is required. This has been an active research focus. Ingrand and Ghallab (2017) survey work on discrepancy detection. Useful to this thread of research, (Molineaux, et al. 2010) discusses it in the context of goal reasoning.

In terms of reasoning models, (Roberts et al. 2018) discuss some interesting concepts in relation to Goal Networks. A life cycle for goals is also discussed. Combining these

goal networks and the temporal nature of goal life cycles, a goal timeline develops. This timeline structure can represent the reasoning needed for some of the explanation models once discrepancies have been detected.

Utilizing models of the world that are not in the explainer's original model is both challenging and novel to pursue for several reasons. It requires an agent to distinguish between different viewpoints of explanations. Introduces reasoning over viable explanations that can be generated. Requires a conversation concerning the ethics of deciding when an agent can decide to deceive. Finally, it opens up the area of XAI to new sets of scenarios - namely those that are deceptive or rebellious.

# 6 Evaluation

To facilitate the development of deceptive or rebellious explanations, we will need a way to evaluate them. We propose a few areas that may be suitable for such testing. One such testing ground is the RoboCup Rescue. This is a popular disaster simulation (Kitano and Tadokoro 2001) that can be leveraged to simulate examples similar to those given in Section 3. Various games and game simulations may prove useful to test for these explanations. Some game options include Minecraft, One Night Ultimate Werewolf, Secret Hitler, Clue, and Diplomacy. Other relevant domains may include those that involve unmanned air and underwater vehicles. These vehicles require a large amount of autonomy and can be utilized in areas where discrepancies between an operator's situation awareness and the vehicle's belief state differ dramatically.

Along with testing simulations, we can also look at measures of explanation effectiveness. Some of these measures can include clarity, timeliness, or correctness. Did the explanation answer the query? How easy was the explanation to understand? Was the time it took to respond seen as adequate? Is the user's attitude toward the agent lower or higher given this form of explanation?

**Acknowledgements** This research was performed while the first author held an NRC Research Associateship award at NRL. Thanks to DARPA and NRL for supporting this research. The views, opinions and/or findings expressed are those of the authors and should not be interpreted as representing the official views or policies of the Department of Defense or the U.S. Government.

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
