# OpenReview forum: "When Agents Talk Back: Rebellious Explanations"
_icaps-conference.org/ICAPS/2019/Workshop/XAIP — XAIP 2019_

### Official Review · AnonReviewer2 · 2019-05-09
**Weak Reject**

**Rating:** 2
**Confidence:** 2

**Review:**

The paper is written in the form of a position paper where authors raise the problems of deceptive or rebellious explanations. Authors give a brief overview of relevant studies and provide examples of potentially deceptive and rebellious explanations, and give some directions for ways of enabling rebellious explanations and providing evaluation.

The topic and the presented research problems are of high interest and significance for XAI community. However, my main concern is the maturity of the paper.  The examples of rebellious explanations described in the paper could be formalized, and there could be given more details to more precisely identify the problems. Additionally, authors identify the ways for generating deceptive explanations by exploiting existing work such as model reconciliation, discrepancy detection or Goal Networks. However, they do not give any clear ideas and suggestions on how any of these methods could be used. Adding at least one example with the formalization of the problem and explanation with the detailed insight on how to utilize the existing approach or proposing a new one would be desired.

---

### Official Review · AnonReviewer1 · 2019-05-09
**A position paper that introduces an important topic, but lacks detail**

**Rating:** 2
**Confidence:** 2

**Review:**

The paper describes the problem of providing rebellious or deceptive explanations. The paper is a position paper that describes the avenue of research and describes the existing related work. The problem is interesting and very relevant to the workshop. The detail of the paper is very light. While a section does list enablers of rebellious explanations, it would be a more useful roadmap with a little more detail on how the various works could be used, what sub-problems can be solved, and how they relate to the examples of deceptive and rebellious explanations in the previous sections.

The section on evaluation is appreciated, but also does not go into enough detail on how rebellious and deceptive agents might be evaluated. A number of game options and robotics domains are listed, but why are these games suitable for this evaluation -- what key features do they possess? Do they match the examples in the previous sections? While I would not expect a position paper to necessarily lay out a detailed scheme for evaluation, more is required here. The authors have in mind how these scenarios can be used, and what might constitute a successful deceptive explanation, but it is not explained.

There is one incorrect citation: the maxims of dishonesty were introduced in the paper:
Chiaki Sakama, Martin Caminada, Andreas Herzig: A formal account of dishonesty. Logic Journal of the IGPL 23(2): 259-294 (2015)
as opposed to:
Chiaki Sakama: A Formal Account of Deception. AAAI Fall Symposia 2015: 34-41

The acronym BS is used without explaining what it means - its first occurrence should be in full. It's a common English phrase, but immediately understood by a non-native speaker.

---

> ### Comment · AnonReviewer1 · 2019-05-20
> **correction**
>
> "but NOT immediately understood by a non-native speaker."
>
> Apologies for any confusion.

---

### Public Comment · ~stephannie_Baker1 · 2019-07-04
**Thank you so much.**

Well, I saw this thing on other websites also but Wikipedia is saying something else about this.

https://en.wikipedia.org/wiki/Rebellion

---

### Decision · Program_Chairs · 2019-05-15

**Decision:**

Accept

**Comment:**

While the reviewers view this paper critically, in the spirit of making the workshop a venue for discussion and feedback we decided to reject only those papers with strong reject votes.

Please address all review criticism as best possible for the final paper version and its presentation at the workshop. Looking forward to discuss your work at the workshop!